# Enablers and Barriers of a Cross-Cultural Geriatric Education Distance Training Programme: The Singapore-Uganda Experience

**DOI:** 10.3390/geriatrics5040061

**Published:** 2020-09-25

**Authors:** Ngoc Huong Lien Ha, Xin Ying Chua, Shallon Musimenta, Edith Akankwasa, Nongluck Pussayapibul, Hui Jin Toh, Mimaika Luluina Ginting, Dujeepa D. Samarasekera, Wai Jia Tam, Philip Lin Kiat Yap, James Alvin Yiew Hock Low

**Affiliations:** 1Yishun Central 2, Geriatric Education and Research Institute, Singapore 768024, Singapore; lien.ha.nh@geri.com.sg (N.H.L.H.); xinying.chua1@gmail.com (X.Y.C.); pussayapibul.nongluck@geri.com.sg (N.P.); ginting.mimaika.luluina@geri.com.sg (M.L.G.); yap.philip.lk@ktph.com.sg (P.L.K.Y.); 2Mildmay Uganda, Kampala, Uganda; smusimenta040@gmail.com (S.M.); edith.akankwasa@mihs.ac.ug (E.A.); 3Department of Geriatric Medicine, 90 Yishun Central, Khoo Teck Puat Hospital, Singapore 768828, Singapore; medthj@nus.edu.sg (T.H.J.); 4Yong Loo Lin School of Medicine, Center for Medical Education, National University of Singapore, Singapore 119077, Singapore; dujeepas@gmail.com (D.D.S)

**Keywords:** videoconferencing, cultural competency, geriatric education, distance learning, health professional education

## Abstract

Background: By 2050, 80% of the world’s older population will reside in developing countries. There is a need for culturally appropriate training programs to increase awareness of eldercare issues, promote knowledge of how to better allocate resources to geriatric services, and promulgate elder-friendly policies. A monthly distance geriatric education programme between a public hospital in Singapore and health institute in Uganda was implemented. This study explored the enablers and barriers to the delivery of culturally appropriate geriatric education programmes via a videoconferencing platform. Methods: We conducted 12 in-depth interviews with six teachers from Singapore and six learners from Uganda. The interviews were audio-recorded, transcribed and analyzed using an inductive thematic approach to analysis with the aid of the NVivo software. Results: Enablers included inter-personal real-time interactions between teachers and learners whereas misaligned perceptions of cross-cultural differences between Singaporean teachers and Ugandan learners were a barrier. Rapport building, teacher motivation and institutional support were perceived to contribute to the programme’s sustainability. Overall, Ugandan learners perceived that the training improved knowledge, skills, attitude and practice of geriatric care. Participants suggested that future initiatives consider aligning cross-cultural perceptions between partners, conducting a training needs analysis, exploring complementary modes of information dissemination, and allotting time for more interaction, thereby reinforcing mutual sharing. Adequate publicity and appropriate incentivisation may also better sustain the programme. Conclusions: Our findings suggest that cross-cultural training via a videoconferencing platform was feasible. Our results inform planners of future distance educational programmes of how to improve standards of cross-cultural competency and forge promising international partnerships.

## 1. Introduction

Between 2000 and 2015, the world’s number of adults aged 60 years or above increased by around 48%. This number is expected to grow 56% by 2030 and will further double by 2050 [1]. Developing countries are expected to bear the brunt of this “unprecedented phenomenon”. By 2050, almost 80% of the world’s older adults and 71% of the world’s “oldest-old” will live in developing nations [1]. Africa will be among the top three continents to be hardest hit with this ageing phenomenon, consisting of up to 105 million elderly people by 2030 and 205 million by 2050, just after Latin America and Asia [1].

At present, developing countries are ill-equipped to deal with these emerging demographics [2]. Most countries in Africa have limited access to geriatric care. International aid agencies may not provide culturally appropriate or relevant services suited to the local context [3]. Uganda, a nation in Sub-Saharan Africa, faces such a situation. Typically, older adults live in the community and lack access to geriatric services. There is no centralised body to coordinate the multi-lateral efforts for the delivery of such care [3]. The problems of older people who live isolated in rural areas are compounded by the young migrating to the cities [3]. The stigmatisation of older adults with medical illnesses [3] underscores the gaps in awareness and geriatric education at the community level. While Uganda’s five-year National Action Plan for Older Persons envisioned to build a secure and dignified environment for the elderly to live in [4], it has not fully translated into elderly-friendly policies, leaving older adults in a state of physical and social vulnerability [3].

With shared goals, a partnership was established between the Geriatric Medicine Department (GRM) of a public hospital in Singapore and a non-governmental health institute in Uganda. This materialized through the collaborative work of one of the authors who had spent two years as a volunteer public health doctor with the latter. A distance training programme utilizing videoconferencing equipment was developed to deliver geriatric education to healthcare professionals in Uganda. The goals were to foster greater awareness of eldercare issues, impart relevant geriatric knowledge and skills and empower healthcare workers to look after older persons in Uganda. The curriculum was co-designed by Ugandan leaders of the health institute and several senior consultants of GRM. Informed by needs assessment feedback from various stakeholders in Uganda and a literature review, the curriculum was designed with a focus to address the gaps in geriatric care in healthcare delivery to older persons among healthcare and community workers in Uganda. A brief outline of the modules taught in the programme is provided in Appendix A. Delivery of content was made via a videoconferencing system. This programme was funded in-kind by both organisations.

Transnational and cross-cultural partnerships in training that utilises online learning platforms as well as teleconferencing are becoming more common with the advancement of technology [5,6,7,8]. In building such collaborations, greater emphasis is placed on cultural competency training and the development of intercultural sensitivity [9]. However, there are attendant difficulties, such as the adaptation of teaching materials from one country to suit the other’s culture and practice, limited opportunities for face-to-face interactions, inadequate facilities and financial cost for infrastructure maintenance [10]. There is a paucity of research investigating the influence of socio-cultural factors as barriers or enablers on effective teaching and learning using online distance modalities in health professional education. This study therefore provides a means to evaluate a cross-cultural distance learning programme between two countries with vastly different socio-demographic and economic backgrounds. It also identifies factors influencing conducive online teaching-learning environments and the related communication processes, which impact programme quality.

This study endeavours to examine the challenges and opportunities of delivering distance training to healthcare professionals of a developing country, via a videoconferencing platform and consequently illuminates the possible ways for the delivery of culturally-competent teaching between countries.

## 2. Materials and Methods

### 2.1. Participants

Out of ten lecturers from Singapore and thirty attendees from Uganda, six participants, through a purposive and convenience sampling approach, were entered into the study to take part in semi-structured in-depth interviews. Participants included in the study were individuals whom played an active role in attending, conducting or coordinating the distance training programme. From Singapore, participants were five geriatricians who conducted at least one lecture and a programme coordinator. From Uganda, participants were healthcare professionals such as clinical officers, allied health staff and a spiritual leader. The participants’ diverse experiences in learning, teaching and coordinating the programme shared from different socio-cultural backgrounds might add valuable insights to the study. As all participants were fully proficient in speaking English, the interviews were conducted using the English language either in person or via teleconferencing from July to December 2016.

### 2.2. Procedures

Eleven monthly sessions were conducted via a videoconferencing system from July 2015 to May 2016, with an average attendance rate of 30 per session. The sessions were conducted live through a broadband line between Singapore and Uganda. A proprietary videoconferencing hardware and software was used for the purposes of the teaching sessions. The time difference between two countries was about five hours. Hence, the most suitable period for the live teaching sessions was in the post-lunch period for Singapore which corresponded to the morning period in Uganda, i.e., 2 p.m. and 9 a.m., respectively. The training involved didactic lectures which covered topics such as geriatric assessment and management, complex care of older persons, end-of-life care and ethical issues. For each session, participants were given a pre- and post-lecture quiz to assess their knowledge of the topic, a satisfaction survey and feedback form for subsequent lectures.

### 2.3. Technology

The hardware that was used for the programme was a videoconferencing system comprising a remote-controlled camera and speakerphone on the Singapore side, and a webcam and laptop with its internal microphone/speaker system on the Uganda side. Connection was made through Wi-Fi and the internet with connection speeds of up to 10 MB/s. The teaching software that we used was a free video-conferencing software application that is available free on the internet.

### 2.4. Study Design

We employed qualitative techniques using a constructivist approach and Grounded Theory to elicit the perspectives of key informants. An interview guide was developed for the Singapore participants (Appendix B). This was adapted for the Ugandan participants to allow for triangulation. The interview guides were used to understand the teaching experience of participants from Singapore, as well as the learning experience of Ugandan healthcare professionals. Ethics approval was obtained from the Domains Specific Review Board, National Healthcare Group, Singapore (Reference number 2015/01128) and Mildmay Uganda Research Ethics Committee (RECREF 0507-2016). All interviews were audio-recorded and transcribed verbatim. The duration of each interview was between 60 to 80 min.

### 2.5. Analysis

An inductive approach to thematic analysis was used to identify key emergent themes. Two members of the team listened to the audio-recordings and scrutinised the transcripts closely in order to gain an overview of the data. They then coded and analyzed the transcripts independently with the use of the NVivo 11 software. Utilising an exploratory approach, nodes and comments were derived for the data in the process of obtaining a preliminary interpretation, deriving meaning and drawing out emergent themes. Codes were developed to represent the identified themes and applied or linked to raw data as summary markers. The list of emergent themes was then organised. In the process of analysis, the researchers were conscious not to impose their own beliefs while interpreting the data and reflected regularly among each other. The entire study team then rigorously reviewed all codes to re-formulate and interpret the themes and sub-themes, making reference to a priori themes from existing literature. Data saturation was reasonably achieved at the fifth transcript shared by the teachers and the sixth transcript shared by the learners.

## 3. Results

### 3.1. Benefits of the Cross-Cultural Distance Learning Programme

#### 3.1.1. Impact on the Ugandan Learners and Their Community

All the Ugandan healthcare professionals perceived significant improvement in their knowledge, skills, attitude and practice towards geriatric care. The programme enhanced their knowledge of falls, depression, dementia, hearing impairment, end of life care and ethical dilemmas. A majority of the healthcare workers felt that the training helped them conduct geriatric assessments more accurately, critically and holistically. In particular, a nurse was motivated to redesign existing assessment forms to better identify her older patients’ needs. Most participants appreciated the training on communication and counseling skills, which helped build greater rapport with patients. Nursing and allied health staff found it helpful to acquire skills to conduct family meetings in an empathetic way and in teaching caregivers. A few participants gained more confidence to share the knowledge learned with other colleagues and their own family members. One clinician felt that the programme helped him treat older persons more appropriately while the skills acquired were also applicable for the non-geriatric populations.

After the completion of the programme, the Ugandan health institute set up a specialist geriatric clinic to cater to the needs of older adults aged 55 and above, which has been utilised by approximately 400 older adults. Apart from assessment and treatment, the clinic delivered training on self-care, nutrition and geriatric symptoms, as well as supported patients with social and economic issues. This initiative ensured a proper referral system and improved care for patients at the health institute. A snowball effect was also noted as Ugandan staff was required to provide training for new staff and disseminate knowledge learnt to local community leaders with the aim of strengthening capacity in long-term care services.

#### 3.1.2. Impact on Teachers

Most teachers perceived that the training programme was novel and worthy to pursue. As they had little knowledge of and experience with Ugandan learners, most expressed a sense of hopefulness that *“a lot can be done”* (S01). However, some teachers cautioned that the impacts were short-termed and were unsure how impactful this engagement had made on Ugandan learners.
“I’m afraid not much (…) because of the short duration we are talking about a 1-h interaction. You kind of say, yah wow I managed to teach or share with somebody half way across the world. But in terms of lasting effect may not be so much because it’s a very short interaction”. (S05)

### 3.2. Enablers and Barriers to the Programme’s Implementation and Delivery

We identified four salient themes pertaining to the barriers and enablers to delivery and implementation of this cross-cultural distance training programme (Table 1).

#### 3.2.1. Impact of Technological Intervention to Facilitate Distance Learning

Several learners expressed satisfaction with the level of interaction during classes via the use of video-conferencing. Videoconferencing overcame extensive physical distances between two countries, and provided opportunities for real-time feedback and interactions for students to raise questions.
“Because we had time to ask questions and at the same time, they could hear and answer our questions. I think it was interactive.” (U05)
“So we gather in the class, we gather in the class and then they set up their computer and they allow the speakers and we see the person who is lecturing there on the Skype and we are given registration form and to ask us what we have learnt, what, what and also, they also do a registration on that we follow up whatever the person from Singapore is teaching us. They ask us questions and we answer and also we encourage, and interact the session most of the time. And it is very interesting and then they show us the slides and the topic, they teach us the topics showing us the slides and if possible we can take notes.” (U02)

However, some teachers opined that videoconferencing restricted non-verbal communications between the learners and themselves, which they identified as a crucial element in the teaching process. Likewise, the ‘personal touch’ was not well-supported by this mode of delivery.
“I would have preferred training sessions where I can see you, touch you and actually sense the body, the language, the tone, the eye movement and all that leh (sic). Then that connection is definitely better.” (S05) “You don’t get the feel of how your class is. Because a lot of times when you teach something, you kind of have to pitch it to the class and it’s an ongoing thing whether they are getting bored or whether they are listening attentively…(S03)
“You find that with the E-learning, they lack personal touch with the facilitator and some people shy away and they don’t ask questions.” [U01]

In addition, a teacher who prefers to conduct classes with group-based sharing and discussions also expressed dissatisfaction with the level of interactions. He noted, however, that his preference using “a workshop style” delivery would most likely be impeded by connectivity problems if using videoconferencing.
“I prefer a workshop style. And ideally what I normally do for ethics right when I deliver eh, ethics teaching I would deliver, I would divide the class up into several groups and then I will give them all a scenario and then each group will discuss. Then after that I will say group 1, what do you’ll think of this scenario. Can you, answer the question—I will give you questions to answer. So what do you think, what are your thought of this particular problem and everybody will share. This is the way I teach, ethics usually. It’s not a lecture style. And how can you do that, on-line over skype. You know how difficult to do that or not?” (S04)

#### 3.2.2. Impact of Cross-Cultural Differences in Teaching and Learning

Most teachers highlighted the importance of context and culture in teaching and learning. They concurred that prior understanding and knowledge of learners’ culture would allow teachers to tailor the lesson content and ensure its relevance to learners. Hence, a few teachers conducted brief research on their target audience’s culture to contextualise their teaching materials.
“Because I thought… to make it relevant to them…, you need to use the drugs that they are familiar with. There’s no point talking about tons about methadone if it’s not available in their country.” (S02)
“I just, I actually just use my usual slides then look for some specific things that are common in Uganda, for example pain, grief, how they do their funeral. Just to understand a little bit more.” (S06)

Similarly, teachers also expressed concerns about whether the knowledge shared was applicable in the learners’ context, i.e., that the knowledge *“may not be that directly transplantable” (S03)*.

On the contrary, several learners were aware of cultural differences and potential incompatibility of the lesson contents. They felt, however, that these differences might not exert a substantial impact on the quality of the course. They attempted to self-adapt the lesson contents to their learning styles and local settings.
“The examples are of course okay but we need to do quite a lot as far as our nation is concerned to keep to the standards that tally with the examples you are giving us. We feel those are the actual examples but in a way they are not applicable in our country” (U03)

#### 3.2.3. Impact of Rapport and Familiarity on Implementation and Delivery

Several teachers reported that it was “difficult to engage” the learners due to the lack of familiarity with one another. One teacher articulated that, because of a lack of familiarity between the two groups, “there is always this barrier” between teachers and learners. Learners reported a lack of rapport between teachers and students. This could be mitigated by gradual and further understanding of the learners to improve the teaching and learning process.
“It would have been good if I had over time, had more encounters with them with a good sense of where they are practicing and therefore to (better) tailor (the course) to them.” (S03)
“If let’s say I am the same person who teach them over a series of maybe 4 to 5 weeks, the same group, then I get to know them and they get to know me. We’re a bit more comfortable. That may be easier.” (S06)

One teacher noted that in the presence of a course coordinator familiar with the learners, *“the (learners) talked a little bit more which makes sense because (the coordinator) knows them personally.” (S06).* This emphasizes the influence of personal relationships in facilitating interactions between the teachers and learners.

#### 3.2.4. Impact of Intrinsic Motivations and External Support on Implementation and Delivery

Apart from enhancing interactions for programme delivery, personal relationships were also reported to exert positive influence on programme implementation. One teacher articulated the need for “emotional bonds” and “connectedness” to motivate his participation in the programme.
“We need emotional bonds, need reasons. So if it was somewhere I had really good friends with and I know their struggles then I would probably be more inclined to (continue teaching).” (S02)

Intrinsic motivations, such as personal values and beliefs, drive the willingness to contribute to the programme as opposed to extrinsic motivations, such as external rewards, which do not aid in the sustainability of the programme.
“I think you just need the will, you need people who want to do it, you need people who see the need for it.” (S02)
“There has to be a vision that this is what we’re hoping to achieve together. So sustainability is not a question, sustainability is a commitment.” (S06)
I think you just have to appeal to the goodness of his heart, that’s all. It must be out of the goodness of his heart, his charitable spirit lah I guess, to want to do this kind of thing.” (S04)

In addition to the factors at the individual and interpersonal level, organisational factors also play a major role in maintaining programme implementation and sustainability. The philosophy and priorities of the organisation influence attitudes towards the programme. Organisational motivation to improve and readiness to change drive the course implementation process and marketing of the programme to the public.
“As an organization, (…) we create time for capacity building… for those classes.” (U01)

Participants also stated that the implementation and sustainability of the programme is aided by “supportive departments” and is very much “relationship-driven”. The presence of good relationships between the course coordinators and higher management enhanced the ease of programme implementation.

## 4. Discussion

Our results showed that a long distance transcultural training programme utilizing videoconferencing was feasible and helped enhanced the knowledge, skills, attitude and practice of Ugandan healthcare professionals in eldercare. This study also highlighted the barriers and enablers to such a training programme. The barriers included misaligned perceptions between teachers and learners while rapport and familiarity with instructors, intrinsic motivations and institutional support were important to foster reciprocal relationships, which contributed to sustainability of the programme.

The first dominant theme evident in the study’s results was the misaligned perceptions between teachers and learners, in terms of interactions and cultural relevance of teaching content. Congruence in perceptions between teachers and learners has been repeatedly demonstrated to be of vital importance in any training programme [11,12]. It allows teachers to attune their teaching styles to the learner’s behaviour in the learning environment, thus influencing the quality of learning as well as the motivations of learners [11,12]. The Singaporean teachers’ perceived lack of interactions as well as culturally relevant guidelines to ensure effective teaching were in contrast to the Ugandan learners who were satisfied with the interactions, were aware of the cultural differences and sought to adapt the lessons to their local settings. This cultural difference could have been mitigated if the trainers had been put through some form of familiarisation or orientation exercise which could raise their cultural competence and allow their teaching to be more effective. These findings could be explained by the different culture and educational approaches of the two countries.

Firstly, the Uganda education system may be one that is more didactic and instructional. Past literature has revealed that learners in an instructional setting tended to focus on passive reception of information and abrogated the responsibility of learning to the teachers [11,13,14]. On the contrary, the Singapore education system has in recent years adopted a more constructivist and interactive approach [15,16], which requires collaboration and inputs from learners [17]. The use of this approach was evident from the interviews with Singapore teachers who were keen to conduct *“on-going discussions” (S03), “workshop style” (S04)* and use methods which were able to *“spark interactions” (S05)*. This collaborative approach also involves learning in social and cultural contexts [17]. This was born out by the interviews whereby teachers asked learners about their *“settings”* and *“cultural beliefs” (S03)*. We believe that the use of the videoconferencing system may have accentuated the views of the teachers who preferred a more interactive approach to teaching but may have had minimal effect on the learners who were more used to a didactic approach to learning.

Secondly, the demographic profile, socioeconomic status and healthcare landscape of Uganda and Singapore had extensive influence on the level of geriatric care in the two countries. Uganda as a developing country with a relatively young population [18], has focused on public health and communicable diseases, whereas geriatric care has been relegated to a lower priority [19,20]. Conversely, Singapore’s population is ageing dramatically and has prompted the initiation of policies targeted at enhancing eldercare and enhanced teaching of geriatrics in the medical curriculum since 2000 [21,22,23]. Differences in levels of priority assigned to geriatric care may account for the different perceptions between the Singaporean teachers and Ugandan learners as well. As the field of geriatrics is still in its infancy in Africa, Ugandan learners may have been more focused on acquiring basic knowledge of geriatric care [24,25]. Singapore, however, being well-established in the field and having a multi-cultural society, has placed much emphasis on cultural competence in medical education, and teachers there tended to take a more collaborative approach. The dissimilar understanding of course objectives may potentially explain the differences in the teaching and learning approaches taken, satisfaction with the level of interactions and perception of the relevance of cultural influences.

Another dominant theme in this study revolves around the sociocultural concept of relationship and network building, which is vital to the success of the programme being studied here. Our results highlight the need for building a network of relationships between teachers and learners as a basis for forging a distance learning programme. This is further helped with the support of the respective sponsoring organisation. Teachers acknowledged the need to be acquainted with and have a personal relationship with learners despite the distance. Past literature has demonstrated the teaching process to be rapport-intensive and rapport has been shown to enhance social interactions, reduce anxiety as well as create a supportive environment where students feel valued and respected [26,27]. The presence of rapport will therefore act as an enabler for the learning process. This study also suggested that rapport may help align perceptions between teachers and learners and prior relationship with the learners may allow teachers to elicit needs and ensure relevance of teaching contents for learners.

Concomitantly, teachers and learners expressed the need for organizational support and meaningful relationships with the administration and management teams of each organisation. Previous studies evaluating cross-distance online learning also suggested that organisational support and capacity are essential in the implementation and sustainability of a programme [5,10]. Inter-organisatinal rapport as well as relationship and network building are also essential factors in programme initiation and sustainability [8].

We submit that, in order, to enhance the delivery of similar cross-cultural distance training programmes, one could align perceptions between teachers and learners as well as conduct a needs analysis to ensure relevance of the training curriculum, prior to commencement of the programme. Future programmes could also consider rapport building, alternative modes of lesson delivery that enhances interactions and provide opportunities for real time bidirectional sharing of knowledge and information (Table 2).

Cross-cultural differences between teachers and learners can be bridged by adopting a culturally inclusive teaching approach [6]. The prerequisite is for teachers to understand cultural similarities as well as differences and the nuances between the two. This could be achieved via a pre-course briefing or orientation session where teachers and learners attempt to understand each other’s needs and expectations in the respective sociocultural context. Learners may also benefit from understanding the instructors’ settings and perspectives [14]. Apart from verbal sharing, knowledge of each country’s cultures may be obtained electronically online or via the print media. Once alignment of perceptions and expectations are achieved, only then can contextualisation of teaching content be made [14]. This may involve in-depth discussions with relevant stakeholders such as programme managers, administrators, teachers and learners, which helps to determine the appropriate learning materials, preferred learning method and modes of teaching delivery to employ [12]. Such discussions should be iterative throughout the programme, so that timely and appropriate changes can be made along the way.

Interpersonal rapport has been consistently highlighted to be an essential element in a distance training programme. This facilitates the teachers’ and learners’ understanding of each other’s learning needs and styles as well as the setting up of an adequate learning environment. Rapport and relationship can be built over time via more frequent interactions. Opportunities for interactions can also be enhanced by, for example, segregating the training into ‘didactic’ and interactive components. The didactic information providing component may be replaced by uploading reading materials i.e., slides and pre-recorded video-lectures onto an online portal. This allows for a greater portion of the videoconferencing time for interactions between teachers and learners. The interactive sessions can then incorporate routine question and answer sessions, sharing of practical wisdom, the use of case studies, role playing, debates in the teaching process. The online platform mentioned also allow learners to leave queries for the teachers at any convenient time, serving as a continuous communication platform between teachers and learners, enabling them to interact more frequently and creatively [13]. More importantly, this approach may mimic a flipped classroom, in which students depend less on teachers’ didactic instructions but are guided to undertake their own self-learning and take ownership of their intellectual stimulation and education [6,13]. In a cross-cultural learning environment, students’ roles should be to move away from being passive recipients of knowledge content to becoming active learners in the educational experience. Lastly, pre-recorded lectures may be used as a structured tools to train different batches of participants at varying periods of the curriculum, thus increasing collaboration and contributing to the sustainability of programmes.

Besides aligning perceptions and building rapport between teachers and learners to ensure the adequacy of training content and delivery, one of the key points of a distance programme is ensuring long term sustainability [14]. These may be achieved via an “all teach all learn” approach. The evidence showed that learning has to be a bidirectional process [8,14]. These allow for mutual learning between partners, as well as improving students’ abilities to present and defend their views clearly, thus enhancing their communication and critical thinking skills [6]. Teachers may also find it helpful to keep an open mind and learn from learners’ unique predispositions and constraints. Opportunities should be provided for mutual learning between partners in future programmes, with the incorporation of local case studies and scenarios. Some of the Singaporean lecturers found it valuable to not only offer knowledge but to learn as well about geriatric care in other countries. All the Ugandan participants also considered this partnership beneficial for mutual sharing and learning and looked forward to its continuation in the future. Future programmes may consider ensuring mutual learning and sharing of information in efforts to maintain sustainability of the programmes [14]. In future programmes, Singapore lecturers may also share standards of geriatric care around the world for Ugandan participants to aspire to.

Due to the practical constraints associated with recruiting overseas participants, the study had a small number of participants. Nonetheless, as the team had gone through rigorous discussion to ensure the perspectives shared had reached a level of consistency, saturation was reasonably achieved with twelve participants. While the teleconferencing platform allowed us to bridge the long distances, attendant technical issues might have limited the clarity of information shared by the participants.

## 5. Conclusions

To our knowledge, this is the first ever exploratory study that evaluated the enablers and barriers in the delivery of a cross-cultural distance training programme in eldercare that straddles two vast continents. Our findings underscored important aspects of pedagogical principles such as knowledge synthesis, dissemination, exchange and application. They highlighted opportunities and challenges of transboundary collaboration among health profession using technological interventions. Our findings suggested that cross-cultural training via a videoconferencing platform was feasible and effective. We also identified critical success factors, which included organisational commitment, teachers’ and learners’ motivations, good communication and culturally relevant pedagogical content. These are essential for the development of future educational programmes, improvement of cross-cultural competency standards and forging promising international partnerships.

## Figures and Tables

**Table 1 geriatrics-05-00061-t001:** Enablers and barriers to tele-teaching programme’s implementation and delivery.

**1. Impact of Technological Intervention to Facilitate Distance Learning**
Learners were engaged by videoconferencing, which allowed opportunities for interactions.
Learners were engaged by videoconferencing, which allowed opportunities for interactions.
Teachers felt that videoconferencing limited interactions such as non-verbal communications.
**2. Impact of Cross-Cultural Differences in Teaching and Learning.**
Teachers emphasized importance of context and culture in teaching.
Learners adapt teaching contents to their local culture and healthcare landscape.
**3. Impact of Rapport and Familiarity on Implementation and Delivery**
Presence of personal relationship with learners facilitates rapport building.
Rapport ensures a suitable learning environment for learners.
Rapport allows teachers to better understand learners’ needs and context.
Continued relationship between teachers and learners build on the effectiveness of teaching.
**4. Impact of Intrinsic Motivations and External Support on Implementation and Delivery**
Teachers’ intrinsic motivations drive the programme’s implementation and sustainability.
External incentives may not drive the programme’s implementation and sustainability.
Teachers’ personal relationships and emotional bonds to learners enhance the programme’s implementation.
Organisational beliefs and capacity enhance implementation and sustainability of the programme.
Presence of rapport and relationships within the department or organization enhances the implementation and sustainability of programme.

**Table 2 geriatrics-05-00061-t002:** Recommendations for future programme’s implementation.

Rationale	Step-by-Step Recommendations
1. Align perceptions and expectations between teachers and learners in order to contextualize teaching contents effectively and create an adequate learning environment.	Organize a pre-briefing session for teachers to understand learners’ culture, context and needs.
	Sharing of each country’s cultures may be documented through books or videos for mutual reference.
2. Discuss with relevant stakeholders to ensure the relevance of course content and delivery.	Conduct needs assessment or discuss with stakeholders to determine the what is relevant to the local context and culture (teaching content, learning style, etc.).
3. Incorporate appropriate modes of lesson delivery to enhance interactive sections of the programme.	Separate course delivery into informational and interactive components.
	Provide pre-readings and pre-recorded lectures as alternative ways to deliver information, so that more time can be dedicated for interaction.
	Provide a platform to allow access to information and provide continuous interactions between teachers and learners.
4. Building rapport between teachers and learners is essential for learning and building further engagement over time.	A single teacher can teach multiple series of a lecture.
	Increase the period of interaction and engagement between teachers and students.
5. Allow for two-way learning whereby both partners are able to learn from each other.	Create opportunities for mutual learning, such as learners sharing their knowledge with the teachers.
	Learners can share their local constraints, innovations and solutions with teachers.
6. Increase the reach of geriatric training programmes to more healthcare workers.	Sensitize and increase the awareness of more healthcare workers to geriatric training programmes.
	Make it compulsory for healthcare staff to attend geriatric training programmes.
	Incentive based motivation—professional recognition and career development.
7. Future operational process.	Increase the lecture time to 1.5 or 2 h.
	Utilize more laptops to allow for smaller group discussions.
	Utilize alternative platform of tele-teaching such as cloud 9.
8. Alternative methods of conducting distance learning programmes.	Continued professional development such as a fellowship programme and study trip to partner countries.

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
