# Peer review of "Enablers and Barriers of a Cross-Cultural Geriatric Education Distance Training Programme: The Singapore-Uganda Experience"

_geriatrics, 2020, doi:10.3390/geriatrics5040061_

Round 1

Reviewer 1 Report

This manuscript addresses an important  topic: how to get effectively get qualified geriatrics instructors, who are usually not co-located with their trainees, to deliver culturally competent geriatrics education and training to health professionals in developing countries.  

Line 20-21: The 2nd sentence of the Abstract is unsubstantiated and should be removed. It could be replaced with a statement such as: There is a need for culturally appropriate training programs to increase awareness of eldercare issues, promote knowledge of how to better allocate  resources to geriatric services, and promulgate elder-friendly policies.  

Line 46: There is a movement to no longer use the term "silver tsunami" because it is a negative term.  Because tsunamis are, in general, frightening and dangerous, the phrase as applied to older adults also implies something frightening and dangerous about aging. This is probably not an image we wish to perpetuate.

Line 95: were the participants chosen randomly or by some specific criteria?

Line 97: should purposively be changed to purposely?

Lines 101-111: The videoconferencing system was not adequately described. If a major conclusion of the study showed that videoconferencing was successful, more details should be given: how was time difference accounted for, were the lectures synchronous aor asynchronous or both; we lectures taped for later review, etc...

Lines 136-155: these results are impressive but I am not sure that they would have been different if the information had been delivered via face-to-face modalities. So the need for the instruction is shown but not the need for videoconferencing.

Line 337: change enhanced to enhance

In the Discussion it is difficult to determine how many of the statements are the result of this study ad how much comes from the literature. This makes it difficult to assess what is new and what is confirmed.

Summary: The paper has too much information. Is the purpose of the paper to show that videoconferencing is an effective teaching tool in developing countries or is it to show that culturally competent instruction is critical and complicated? If the former, then more info is needed about the logistics of the videoconferencing and what the decision points were on how and when and to whom it would be delivered.  if the latter, then more info is needed about the barriers presented by videoconferencing when the teachers and learners are of different socioeconomic and national cultures (not to mention that teaching and learning are different cultures).

Sample sizes were small. A statement about the impact of a small sample size on overall conclusions would be appropriate. Was there any possible bias introduced by the small sample size?   

Author Response

Response to Reviewers’ Comments

On the manuscript ID geriatrics-916093 entitled "Enablers and Barriers of a Cross-cultural Geriatric Education Distance Training Programme: the Singapore-Uganda Experience”

We would like to thank to the reviewer and editors for their thoughtful and constructive comments to help improve the manuscript. Our responses to the comments are appended below, with changes in the manuscript via tracked changes.

Reviewer 1

This manuscript addresses an important  topic: how to get effectively get qualified geriatrics instructors, who are usually not co-located with their trainees, to deliver culturally competent geriatrics education and training to health professionals in developing countries.  

COMMENT 1:

Line 20-21: The 2nd sentence of the Abstract is unsubstantiated and should be removed. It could be replaced with a statement such as: There is a need for culturally appropriate training programs to increase awareness of eldercare issues, promote knowledge of how to better allocate resources to geriatric services, and promulgate elder-friendly policies.

RESPONSE 1: We have made the changes as suggested.  

COMMENT 2:

Line 46: There is a movement to no longer use the term "silver tsunami" because it is a negative term.  Because tsunamis are, in general, frightening and dangerous, the phrase as applied to older adults also implies something frightening and dangerous about aging. This is probably not an image we wish to perpetuate.

RESPONSE 2: We have replaced the phrase “silver tsunami” with “unprecedented phenomenon”

COMMENT 3:

Line 95: were the participants chosen randomly or by some specific criteria?

RESPONSE 3: The participants of the study were chosen from among the students of the teaching programme based on a convenience sampling approach

COMMENT 4:

Line 97: should purposively be changed to purposely?

RESPONSE 4: we took a purposive and convenience sampling approach when recruiting participants into our study. We have modified the sentence at line 105-107 to be like this, ‘…., six participants, through a purposive and convenience sampling approach, were entered into the study to take part in semi-structured in depth interviews’

COMMENT 5:

Lines 101-111: The videoconferencing system was not adequately described. If a major conclusion of the study showed that videoconferencing was successful, more details should be given: how was time difference accounted for, were the lectures synchronous aor asynchronous or both; we lectures taped for later review, etc...

RESPONSE 5: The sessions were conducted live through a broadband line between Singapore and Uganda. A proprietary videoconferencing hardware and software was used for the purposes of the teaching sessions. The time difference between two countries was about five hours. Hence, the most suitable period for the live teaching sessions was in the post-lunch period for Singapore which corresponded to the morning period in Uganda, ie 2pm and 9am, respectively

COMMENT 6:

Lines 136-155: these results are impressive but I am not sure that they would have been different if the information had been delivered via face-to-face modalities. So the need for the instruction is shown but not the need for videoconferencing.

RESPONSE 6: We concede that the content of the teaching in itself was important and there was no real substitute for in-person face-to-face interactive teaching; in addition, the value of the mode of delivery was less certain. However, we believe that the use of IT had bridged the distance between two countries across the continents and circumvented the heavy logistics needs and costs for in-person teaching, especially when the teachers and students come from separate countries and continents.

COMMENT 7:

Line 337: change enhanced to enhance

RESPONSE 7: We have made the changes suggested

COMMENT 8:

In the Discussion it is difficult to determine how many of the statements are the result of this study ad how much comes from the literature. This makes it difficult to assess what is new and what is confirmed.

RESPONSE 8: We provided interpretation of our findings and attempted to explain, theorise and reason out our findings and, where necessary, provide relevant information from the literature to concur or refute the former. This is borne in part by the referencing and references that are provided at the end of the paper. We also offer some suggestions that we think might address the barriers and enhance the facilitators.

COMMENT 9:

Summary: The paper has too much information. Is the purpose of the paper to show that videoconferencing is an effective teaching tool in developing countries or is it to show that culturally competent instruction is critical and complicated? If the former, then more info is needed about the logistics of the videoconferencing and what the decision points were on how and when and to whom it would be delivered.  if the latter, then more info is needed about the barriers presented by videoconferencing when the teachers and learners are of different socioeconomic and national cultures (not to mention that teaching and learning are different cultures).

RESPONSE 9: This was a descriptive, qualitative and exploratory look at a new way of teaching between two very different societies. We looked particularly at the barriers and enablers of such a novel approach to teaching, ie the use of old technology in a new way and how it was able to bridge the physical distance and connect two very different societies in a teaching/learning programme.

We have added some more information on the barriers related to the use of the videoconferencing platform.

Cultural differences featured quite prominently as a barrier to some aspects of the programme and we agree that this is above and beyond the modality or platform employed to teach or learn. The novelty of this study is that it is the only one of its kind that looked at a niche area of transboundary (Africa- Asia) training and learning using IT as a platform or modality.

COMMENT 10:

Sample sizes were small. A statement about the impact of a small sample size on overall conclusions would be appropriate. Was there any possible bias introduced by the small sample size?   

RESPONSE 10: This was a qualitative study of a small pilot project. We felt that we could reach data saturation on the numbers interviewed. We adopted an inductive approach and developed the themes and hypothesis as the data was analysed. The issue of bias does not arise as this is a constructivist approach to building the hypothesis. We concede too that in qualitative studies, the way the data is interpreted is somewhat influenced by the worldview of the researcher(s).

Reviewer 2 Report

The authors address an important issue, given the predicted demographic trends in developing countries in general. It would be helpful to include demographic predictions specifically for Uganda, if available. Distance education on geriatrics provided by experts and targeting developing countries could be effective in assisting countries such as Uganda to prepare for the anticipated demographic shift. The consideration of issues involved in offering such an option is a welcome addition to the field.

  1. It would be helpful to provide a brief history of the origins of this program. Singapore and Uganda are not obvious partners, at least to readers unfamiliar with gerontological efforts in these countries. Who initiated the program? What is the funding source?
  2. Provide more detail on the co-design of the program. Although it was “informed by needs assessment feedback from various stakeholders in Uganda,” one of the final recommendations was to conduct a needs assessment. Which stakeholders participated in the initial needs assessment? Did the needs assessment include input from older adults and their family caregivers, the ultimate consumers of geriatric care, who may be uniquely positioned to identify gaps in the system? An “extensive literature review” also informed development of the curriculum. Did the authors consult international professional guidelines, such as AGHE (Academy for Gerontology in Higher Education) Standards and Guidelines for Gerontology and Geriatrics and the AGHE Gerontology Competencies for Undergraduate and Graduate Education (the latter is available online for free)? Do similar guidelines exist for Singapore and/or Uganda?
  3.  Technology to support distance education has evolved greatly since 2016, the time the program was developed. Dissatisfaction regarding level of interaction may no longer be an issue with the availability of advances such as Zoom, breakout rooms, chat boxes, whiteboards. It would be helpful to readers to provide a brief description of the state of the art at that time and acknowledge that current technology allows the potential for more interactive encounters.
    4. Provide more information on the characteristics of the program learners (e.g., education, prior training and experience working with older adults). How were they recruited?
    5. Pre- and post-lecture quizzes, tapping knowledge of subject matter, satisfaction with the lecture, and open-ended feedback were administered for each lecture, however, results/feedback did not seem to be included in the manuscript.
    6. The instructors’ and the learners’ evaluations of the training experience differed a great deal. Cultural differences in learning styles may help to explain, but this issue should be explored in greater depth. Given the positive outcomes of the training (e.g., redesign of existing evaluation forms, development of a specialist geriatric clinic), one would expect more positive evaluations from the instructors. What outcomes did they anticipate from the training? How did they define success/the course objectives for this program?
    7. Most instructors had little knowledge of or experience with Ugandan learners, in contrast to the statement that they emphasized cultural competence in medical education. Describe the background training the Singapore instructors received, if any, on Ugandan culture and health care systems and beliefs. The recommendation for a pre-course briefing or orientation session may not suffice. There is an extensive literature on cultural competence and ways to acquire it, which should be considered.
    8. The authors should make better use of quotes to illustrate and support their findings.
    9. Was this a one-time event, or did it continue with new cohorts of learners?

Author Response

Response to Reviewers’ Comments

On the manuscript ID geriatrics-916093 entitled "Enablers and Barriers of a Cross-cultural Geriatric Education Distance Training Programme: the Singapore-Uganda Experience”

We would like to thank to the reviewer and editors for their thoughtful and constructive comments to help improve the manuscript. Our responses to the comments are appended below, with changes in the manuscript via tracked changes.

Reviewer 2

The authors address an important issue, given the predicted demographic trends in developing countries in general. It would be helpful to include demographic predictions specifically for Uganda, if available. Distance education on geriatrics provided by experts and targeting developing countries could be effective in assisting countries such as Uganda to prepare for the anticipated demographic shift. The consideration of issues involved in offering such an option is a welcome addition to the field.

COMMENT 1:

It would be helpful to provide a brief history of the origins of this program. Singapore and Uganda are not obvious partners, at least to readers unfamiliar with gerontological efforts in these countries. Who initiated the program? What is the funding source?

RESPONSE 1: We have included the following to the 3rd paragraph of the introduction section of the paper:

Line 70-72- ‘This materialized through the collaborative work of one of the authors who had spent two years as a volunteer public health doctor with the latter.

Line 80-81- ‘This programme was funded in-kind by both organisations’.

COMMENT 2:

Provide more detail on the co-design of the program. Although it was “informed by needs assessment feedback from various stakeholders in Uganda,” one of the final recommendations was to conduct a needs assessment. Which stakeholders participated in the initial needs assessment? Did the needs assessment include input from older adults and their family caregivers, the ultimate consumers of geriatric care, who may be uniquely positioned to identify gaps in the system? An “extensive literature review” also informed development of the curriculum. Did the authors consult international professional guidelines, such as AGHE (Academy for Gerontology in Higher Education) Standards and Guidelines for Gerontology and Geriatrics and the AGHE Gerontology Competencies for Undergraduate and Graduate Education (the latter is available online for free)? Do similar guidelines exist for Singapore and/or Uganda?

RESPONSE 2:

Our healthcare counterparts in Uganda were involved in the initial needs assessment. We concur with the reviewers that ideally, the needs assessment should include inputs from older adults and their family caregivers. We regret that we did not consult any of the  professional guidelines recommended while there are no local guidelines available in Singapore or Uganda. However, we based the curriculum on the teaching experience of WJT and PY (two of the authors of this paper). We conducted a search of the education research literature; however, we concede that the word “extensive” may be rather subjective and relative. Therefore we have decided to remove that word from the main text. Please refer to line 77-78.

COMMENT 3:

Technology to support distance education has evolved greatly since 2016, the time the program was developed. Dissatisfaction regarding level of interaction may no longer be an issue with the availability of advances such as Zoom, breakout rooms, chat boxes, whiteboards. It would be helpful to readers to provide a brief description of the state of the art at that time and acknowledge that current technology allows the potential for more interactive encounters.

RESPONSE 3: We concede that technology has progressed in leaps and bounds over the last 4 years. This is compounded by the pandemic that has forced the digitalisation of many aspects of healthcare and healthcare-related education.

We will include a new subsection (2.3) at Line 127-132, in the Materials and Methods section commenting on the ‘state of the art at that point in time’ and the technology that we used:

2.3 Technology

The hardware that was used for the programme was a videoconferencing system comprising a remote-controlled camera and speakerphone on the Singapore side, and a webcam and laptop with its internal microphone/speaker system on the Uganda side. Connection was made through Wi-Fi and the internet with connection speeds of up to 10 MB/s. The teaching software that we used was a free video-conferencing software application that is available free on the internet.

COMMENT 4:

Provide more information on the characteristics of the program learners (e.g., education, prior training and experience working with older adults). How were they recruited?

RESPONSE 4: Recruitment was carried out based on a convenience and purposive sampling approach. The programme learners were all healthcare professional from various backgrounds such as social work, nursing, medicine, etc. Their education level was secondary school and above. All had existing and prior experience working with older adults. 

COMMENT 5:

Pre- and post-lecture quizzes, tapping knowledge of subject matter, satisfaction with the lecture, and open-ended feedback were administered for each lecture, however, results/feedback did not seem to be included in the manuscript.

RESPONSE 5: We did not include them into the study as we felt that it was beyond the scope of the project. We were not specifically looking at or evaluating the content of the teaching programme. Moreover, the numbers recruited would not have allowed us to derive any meaningful information from the course evaluation perspective (quizzes, MCQs, etc) especially if a quantitative analysis approach was taken. Qualitatively, we have included some satisfaction-related questions into our in-depth interviews. However, we do agree that this could add a new dimension to the study for the future.

COMMENT 6:

The instructors’ and the learners’ evaluations of the training experience differed a great deal. Cultural differences in learning styles may help to explain, but this issue should be explored in greater depth. Given the positive outcomes of the training (e.g., redesign of existing evaluation forms, development of a specialist geriatric clinic), one would expect more positive evaluations from the instructors. What outcomes did they anticipate from the training? How did they define success/the course objectives for this program? 

RESPONSE 6: We did not specifically explore this with the instructors.  This was an oversight as we were more preoccupied to look at the feasibility of conducting this novel teaching modality. Nonetheless, future studies of this nature should look at this important area of teaching as an outcome measure.

COMMENT 7:

Most instructors had little knowledge of or experience with Ugandan learners, in contrast to the statement that they emphasized cultural competence in medical education. Describe the background training the Singapore instructors received, if any, on Ugandan culture and health care systems and beliefs. The recommendation for a pre-course briefing or orientation session may not suffice. There is an extensive literature on cultural competence and ways to acquire it, which should be considered. 

RESPONSE 7: The Singapore trainers were mostly doctors trained in the local universities who took on membership with the Royal College of Physicians of the UK.  Most of them have never taught overseas, let alone in an African country. As this was an opportunistic effort and the teachers were busy clinicians, there was no orientation or pre-course briefing instituted prior to commencement of the programme. We agree, with hindsight, that some form of familiarisation measures should have been implemented prior to the start of the programme. We will include this into the paper as well:

Line 306-309: ‘This cultural difference could have been mitigated if the trainers had been put through some form of familiarisation or orientation exercise which could raise their cultural competence and allow their teaching to be more effective.’

COMMENT 8:

The authors should make better use of quotes to illustrate and support their findings. 

RESPONSE 8: We have included further quotes as requested. This may add to the word count.

COMMENT 9:

Was this a one-time event, or did it continue with new cohorts of learners?

RESPONSE 9: This was a one off pilot project that was not repeated. However, the lessons learnt were translated into another distance-learning programme with Thailand which some of the authors are involved in.

Round 2

Reviewer 1 Report

Your responses are much appreciated.